# Vegetation Analysis of the Area Surrounding a Wild Nest of Stingless Bees *Tetragonula laeviceps* (Smith, 1857) in Sumedang Regency, West Java

Susanti Withaningsih [1,2,3,*] , Valerie Lubay [1], Fakhrur Rozi [2,3] and Parikesit Parikesit [1,2,3]

1 Department of Biology, Faculty of Mathematics and Natural Science, Universitas Padjadjaran, Sumedang 45363, Indonesia; valerie19001@mail.unpad.ac.id (V.L.); parikesit@unpad.ac.id (P.P.)
2 Master Program on Sustainability Science, Graduate School, Universitas Padjadjaran, Jl. Dipati Ukur No. 35, Bandung 40132, Indonesia; fakhrur12001@mail.unpad.ac.id
3 Center for Environment and Sustainability Science, Universitas Padjadjaran, Jl. Sekeloa Selatan No. 1, Bandung 40132, Indonesia
* Correspondence: susanti.withaningsih@unpad.ac.id; Tel.: +62-22-2502176

**Abstract:** Indonesia is a mega biodiversity country that has a very abundant diversity of plants. Plant diversity is inseparable from the role of insects that help to pollinate them. One such insect is the stingless bee *Tetragonula laeviceps*, a bee that has a relatively small-to-medium body size and does not have a sting. Sumedang Regency is one of the regencies in the province of West Java; most of the Sumedang area consists of mountains and has protected forest areas with high biodiversity. The purpose of this study is to determine the biodiversity index of the vegetation around the *Tetragonula laeviceps* wild nests. Data collection is carried out using the method of inventory and systematic checkered lines, with a total plot of 320. The results of the study show that the vegetation is composed of 229 plant species from 75 families. The most dominant vegetation type with the highest value of KR, FR, DR, and INP in all plant categories is the Aren tree species (*Arenga pinnata*). The Shannon–Wiener diversity index in this tree category is 3186, the pole category is 3107, the sapling category is 3418, the seedling category is 3657, and the understory plants category is 3409, with all of the plant categories included in the high-diversity category.

**Keywords:** stingless bee; Sumedang; *Tetragonula laeviceps*; vegetation analysis

## 1. Introduction

Indonesia is a country of mega biodiversity, boasting an extraordinary wealth of diverse plant life within its borders. Additionally, the expansive array of ecosystems encompasses a diverse range from tropical rainforests to coastal mangroves. The ecological diversity present in this particular ecosystem serves as a catalyst for the proliferation of a myriad of plant species, a substantial proportion of which exhibit a strong dependence on insect pollinators for successful reproduction.

The intricate interplay between the botanical realm and the vital process of pollination, wherein insects, most notably the highly valuable bees, play a pivotal role, is inexorably linked to the rich tapestry of plant diversity. Bees, one of the most outstanding creatures of the insect world, are known to develop colonies and are classified within the Apidae family. In the Indonesian archipelago, a frequently encountered group of bees, which are devoid of stinging capabilities, pertains to the taxonomic classification of the genus *Tetragonula*.

Bees, being highly proficient pollinators, play a pivotal role in upholding the equilibrium and vitality of ecosystems. These organisms not only serve the vital function of facilitating the reproductive processes of diverse botanical taxa but also exert a profound influence on the assemblage and relative abundance of coexisting organisms within their respective ecological systems. The network of interdependencies showcased herein underscores the indispensable contribution of bees in upholding and perpetuating biodiversity.

The Apidae family, to which the bees belong, has a diverse array of species, each exhibiting a plethora of unique characteristics and behaviors. In the Indonesian archipelago, the genus *Tetragonula* emerges as a noteworthy assemblage of stingless bees that exhibits notable prevalence. These diminutive organisms demonstrate complex social behaviors, culminating in the formation of colonies characterized by elaborate structures and a sophisticated division of labor among their constituents.

In the vernacular Sundanese dialect, it is customary to designate the stingless bees belonging to the *Tetragonula* genus as "*Teuweul*" [1]. The Sundanese populace has undoubtedly engaged in extensive interactions and astute observations of these bees across numerous generations, thereby augmenting their profound comprehension of indigenous biodiversity. Comprehending the ecological significance of these bees, particularly in the complex process of pollination, holds paramount importance for the advancement of conservation endeavors and the promotion of sustainable agricultural practices in the Indonesian context.

*Tetragonula*, a fascinating genus of bees, exhibits a diminutive to moderate size range and intriguingly lacks the presence of a sting. The distribution of these bees comprises a range of geographically diverse regions, spanning across Australia, Indonesia, New Guinea, Malaysia, Thailand, the Philippines, India, Sri Lanka, and the Solomon Islands. In accordance with conventional knowledge, these insects exhibit a proclivity for establishing enduring colonies, wherein they diligently procure pollen and nectar as their primary means of sustenance. The nesting behavior of these organisms typically involves the construction of subterranean dwellings as well as the utilization of wood cavities, arboreal surfaces, and occasionally even termite mounds as nesting sites and houses that are 1–5 m above ground level [1–3]. Stingless bees have various entrance shapes, such as funnel shapes, irregular circular shapes, or shapes without a protrusion at the entrance [4]. *Tetragonula* bees are cold-blooded animals (poikilotherms) that can survive temperatures around 28–36 °C [5].

The activity of flying out of the nests in search of food and resin sources is related to the growth and reproduction of stingless bee colonies. The success of stingless bees in bringing food sources in the form of nectar and pollen from flowers on surrounding plants affects the productivity and development of the colony [6,7]. The availability and diversity of food sources greatly determine the quality and quantity of bee products such as honey, propolis, and bee bread [8,9].

The source of food for stingless bees comes from almost all types of flowering plants except plants that contain toxic compounds [10]. Plants in forests, plantation areas, or agricultural areas will produce nectar and pollen through the flowering parts of plants which are used as honeycombs by these bees, while honey bees will help with the process of pollinating the flowers [11].

The bees under consideration have exhibited high degrees of adaptability to their respective habitats, effectively capitalizing on a diverse array of nesting sites and resources to sustain their colonies. The nesting behavior exhibited by these creatures, characterized by their propensity to construct nests in a multitude of diverse locations, undoubtedly underscores their profound ecological significance as invaluable pollinators and indispensable agents of biodiversity.

The *Tetragonula laeviceps* bees, indeed, manifest remarkably intriguing social behaviors within the confines of their colonies. The elaborate interplay and allocation of tasks among individuals exemplify the extraordinary adaptability exhibited by these diminutive organisms. Notwithstanding their dearth of a venomous apparatus, these organisms have undergone evolutionary processes to effectively discharge pivotal functions within their respective ecosystems, primarily by facilitating the crucial process of plant reproduction through the mechanism of pollination.

In Indonesia, one of the regions with a significant population of *Tetragonula laeviceps* bees is located in the Sumedang Regency. The Sumedang Regency is situated in the West Java Province. The majority of Sumedang's area consists of mountains and includes pro-

tected forest areas with high biodiversity. Based on information from the local community, *Tetragonula laeviceps* bees are commonly found in forested areas and are classified as wild bees [12].

In a recent study carried out by [13], it was discovered that the Cimalaka subdistrict, situated in the Sumedang Regency, possesses a diverse array of vegetation. This particular area is home to approximately 2334 individual plants, representing 9 distinct species spanning across 5 families. Notably, the lower plant types exhibited a prominent presence, showcasing their ecological significance as both habitats and nourishment for *Tetragonula* bees. Furthermore, the scholarly investigation conducted by [14] elucidated that within the confines of Cilembu Village, situated in the Sumedang Regency, a significant assemblage of 78 distinct species of understory plants were meticulously documented, spanning across an impressive array of 30 taxonomic families. As a distinguished scholar in the field of biodiversity, it is worth noting that the Sumedang Regency is endowed with an abundance of plant species, which holds great promise in terms of providing favorable ecosystems for the *Tetragonula laeviceps* bees.

The interplay between the vast array of plant species holds paramount significance in the tapestry of existence for the *Tetragonula laeviceps* bees. The dearth of vegetation in the vicinity of *Tetragonula laeviceps* bee colonies may potentially engender grave ramifications for their long-term sustenance and viability. Therefore, it is of utmost importance to undertake a comprehensive investigation pertaining to the thorough examination of the botanical composition in the immediate vicinity of the nesting sites inhabited by the indigenous species of stingless bees within the geographical confines of the Sumedang Regency, situated in the province of West Java. This particular set of data holds utmost significance in comprehending the multifaceted dynamics of the flora surrounding the habitats of these stingless hymenopterans and their colonies.

## 2. Materials and Methods

### 2.1. Study Area

The study area, Sumedang Regency is located at $6°51'0''$ south latitude and $107°55'12''$ east longitude. It has a tropical climate, with an average yearly temperature of 24.7 °C and an average rainfall of 2570 mm, with the highest rainfall occurring in December–January [15]. The topography of most areas are hills and mountains with Mount Tampomas being the highest peak (1684 m) and the lowest plateau of 26 m.

Forests cover more than half of Sumedang (about 70%) followed by fields and plantations (17.86%), water and wetlands (7.49%), and residential areas (4, 28%) [16]. This composition of land cover in Sumedang, especially in some sub-districts such as Buah Dua, Cisitu, Darmaraja, and Tanjungkerta, makes it an area with high plant diversity, increasing its potential to support the habitat areas for stingless bees.

### 2.2. Methodology

The methodology employed in this investigation entails the utilization of a survey-based approach. This study was carried out through a series of carefully planned stages, encompassing a preliminary survey, an intensive survey, and a detailed identification process. The initial survey was conducted by gathering data pertaining to the geographical distribution of indigenous *Tetragonula laeviceps* bee colonies within the Sumedang Regency. The comprehensive survey entailed the detailed collection of vegetation data utilizing the quadrant method in the vicinity of the untamed nests.

Each district from Figure 1 had 3 sampling points based on the nests found (see Table 1).

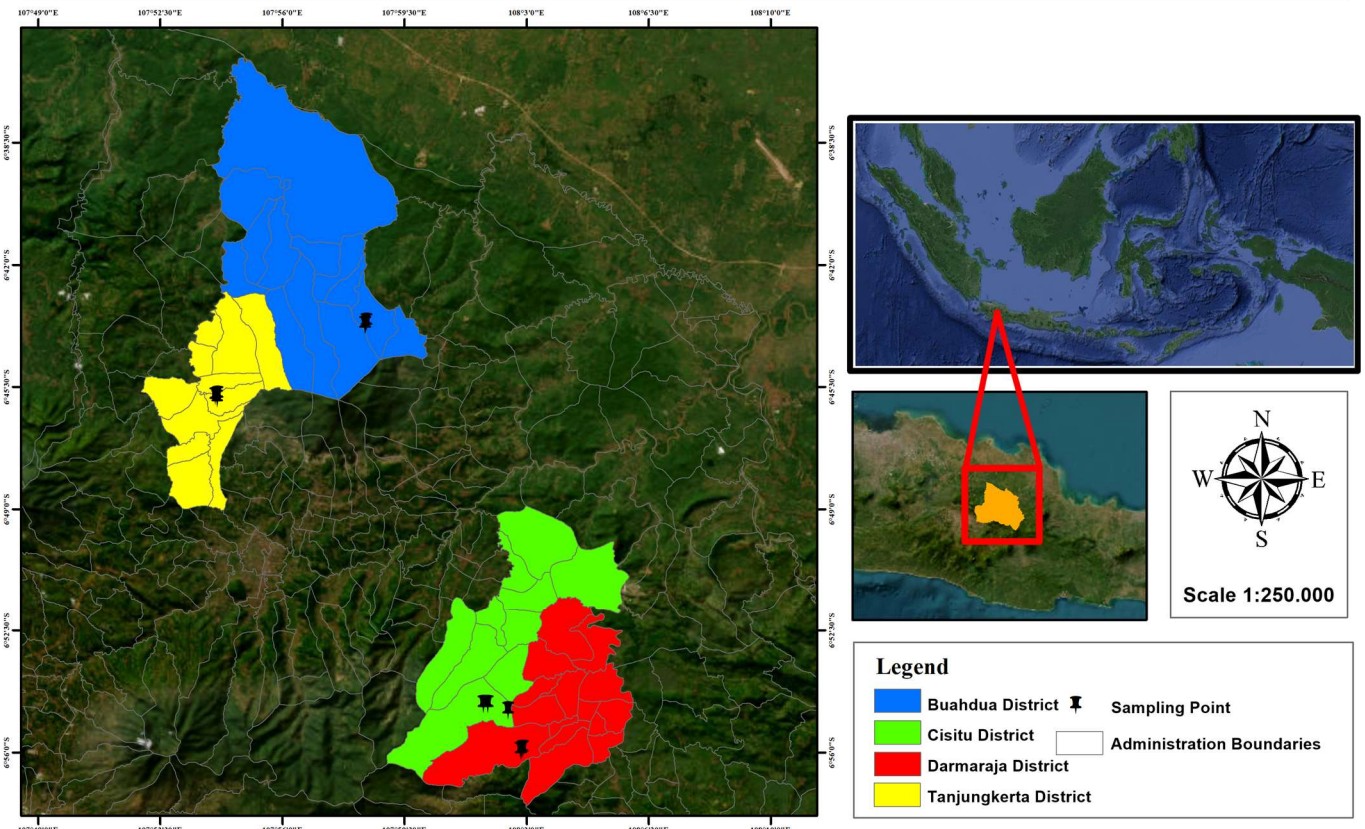

**Figure 1.** Research area.

**Table 1.** Sampling Coordinate Point.

| No | Sub-District | Category | Point | Coordinate |
|---|---|---|---|---|
| 1 | Tanjungkerta | Wild nest | A1 | S6°45′49.4100″ E107°54′08.1800″ |
| 2 | Tanjungkerta | Wild nest | A2 | S6°45′43.4879″ E107°54′08.8242″ |
| 3 | Tanjungkerta | Wild nest | A3 | S6°45′45.6000″ E107°54′04.6500″ |
| 4 | Cisitu | Wild nest | B1 | S6°54′48.1100″ E108°02′28.6100″ |
| 5 | Cisitu | Wild nest | B2 | S6°54′37.2300″ E108°01′53.3700″ |
| 6 | Cisitu | Wild nest | B3 | S6°54′37.3359″ E108°01′46.1346″ |
| 7 | Darmaraja | Wild nest | C1 | S6°55′53.5807″ E108°02′54.1009″ |
| 8 | Darmaraja | Wild nest | C2 * | S6°55′56.1466″ E108°02′51.1511″ |
| 9 | Darmaraja | Wild nest | C3 * | S6°55′56.6490″ E108°02′49.8447″ |
| 10 | Buah Dua | Wild nest | J1 ** | S6°43′42.9615″ E107°58′22.6786″ |
| 11 | Buah Dua | Wild nest | J2 ** | S6°43′42.9965″ E107°58′23.0162″ |
| 12 | Buah Dua | Wild nest | J3 | S6°43′38.1687″ E107°58′24.8238″ |

Noted: * Point C2 and C3 were combined because the distance between these points was >50 m. A transect was made between two points. Additionally, the extremely steep condition made it difficult to collect vegetation data at both points. ** Point J1 and J2 were combined because the distance between these points was >50 m. A transect was made between two points. Additionally, the extremely steep condition made it difficult to collect vegetation data at both points.

In the preliminary phase of sample collection, random purposive sampling was implemented. Transects spanning a length of 100 m were carefully established, taking into account the wide span of the foraging range of the stingless bees (from 53.61 to 162.21 m) [17]. Furthermore, it is worth noting that the maximum distance attainable from the nest, an impressive 497 m, was also considered during the establishment of these transects.

The sampling area covered an area of 3140 square meters, which is a 10% proportion of the overall area of 3.14 hectares, as derived from the circular region with a radius measuring 100 m, which incidentally corresponds to the length of the transect. The transect lines were delineated in four cardinal directions, radiating outward from the epicenter of the nest, thereby establishing a comprehensive spatial framework. The determination of the direction of each transect line was conducted, taking into account the nature of the land cover surrounding the nest point as well as the level of accessibility to the designated transect area.

Following the cardinal directions, a total of ten precise plots were established. Subsequently, a randomization process was employed to select a subset of eight plots from this established pool. In the event that there were plant species that were not encountered within the designated plot, yet were indeed present within the research area, a comprehensive inventory was undertaken with the primary objective of procuring complete information about the entirety of plant species in close proximity to the natural habitats of the indigenous stingless bee colonies.

In this study, the plot sizes were classified based on the plant habitus in the following manner:

- A spatial allocation of $10 \times 10$ m$^2$ was designated for tree species with a diameter exceeding 20 cm.
- For shrub species characterized by a diameter ranging from 10 to 20 cm, an area of $8 \times 8$ m$^2$ was used.
- Sapling species, denoted by a diameter below 10 cm and a height surpassing 1.5 m, were allotted a space of $4 \times 4$ m$^2$.
- Seedling species, distinguished by a height below 1.5 m, were accommodated within a $2 \times 2$-m$^2$ area.
- Ground cover plants, on the other hand, necessitated a smaller allocation of $1 \times 1$ m$^2$. The specific plant species within this category can be determined utilizing the Braun–Blanquet scale, as outlined by [18].

According to the findings of the survey, the sampling points were documented methodically and are conveniently displayed in Table 1. It is worth noting that a substantial number of plots, precisely 320 from the 12 sampling points in 4 districts, were established for the purpose of this study.

The analysis of vegetation was undertaken by taking into account various parameters, specifically density (D) and relative density (RD), frequency (F) and relative frequency (RF), dominance (D) and relative dominance (RD), the importance value index (IVI), and the species diversity index (H').

## 3. Results and Discussion

In the region of Kabupaten Sumedang, a significant proportion of its land, precisely 70.37%, is enveloped by lush and verdant forests. This is followed by areas dedicated to cultivation, including fields and plantations, which account for 17.86% of the region's land. Additionally, it is worth acknowledging the presence of water bodies and wetlands, constituting 7.49% of the area, and settlements, covering 4.28% of the land [16]. The prevailing landscape of Kabupaten Sumedang is characterized by extensive forested regions, plantations, and agricultural activities, thereby signifying a high propensity for harboring a diverse array of plant species. This, in turn, renders the region highly conducive to fostering the natural habitat of the stingless bee species known as *Tetragonula laeviceps*. The domiciles of indigenous *Tetragonula* bees within the jurisdiction of Kabupaten Sumedang are dispersed throughout various sub-districts, specifically Buah Dua, Cisitu, Darmaraja, and Tanjungkerta.

Within each of the aforementioned sub-districts, it is imperative to note that a detailed selection process was undertaken to identify three distinct nest points for the purpose of data collection. In the sub-districts of Darmaraja and Buah Dua, the prevailing land utilization predominantly manifests as secondary forests. As elucidated by [19], the genesis

of secondary forests can be attributed to the anthropogenic intervention of logging activities undertaken by forestry enterprises. The prevalence and constitution of arboreal taxa in secondary woodlands are typically diminished in comparison to their primary, undisturbed counterparts. In the Tanjungkerta sub-district, one observes a prevailing presence of residential settlements. The nesting sites in Tanjungkerta are predominantly situated within the residential compounds. A residential compound, as elucidated by [20], is an expanse of land encompassing a dwelling, meticulously adorned with a diverse array of botanical specimens, be it a solitary species or a harmonious amalgamation of multiple taxa. In the Cisitu sub-district, one can observe a prevalence of mixed gardens as the dominant land use pattern. As per the research conducted by [21], it has been expounded that mixed gardens exhibit a diverse array of plant species, wherein it is imperative for at least one of the constituents to possess a woody nature. The botanical composition within the designated research sites is primarily characterized by the prevalence of arboreal taxa.

### 3.1. Composition and Vegetation Structure around Wild Tetragonula Laeviceps Nest

The comprehension and delineation of the vegetation's composition and structure assume paramount significance in order to facilitate the preservation of stingless bee habitats within the community. According to the findings of our comprehensive investigation, a total of 237 plant species belonging to 75 distinct families have been successfully identified. These species show a diverse array of botanical life, with a total of 8480 individuals being classified into various growth forms, including trees, poles, shrubs, seedlings, and ground vegetation. The analysis of vegetation was conducted by considering various parameters, such as density (D) and relative density (RD), frequency (F) and relative frequency (RF), dominance (D) and relative density (RD), and the importance value index (IVI).

### 3.1.1. Tree

From the careful examination and analysis of the vegetation data in various sub-districts, a total of 52 distinct tree species belonging to 25 diverse families have been successfully identified. Within the realm of arboreal taxonomy, it is noteworthy that the tree classification covers a quintet of preeminent botanical species. These include *Arenga pinnata*, commonly referred to as the Aren palm, the venerable *Tectona grandis*, known colloquially as teak, the illustrious *Ficus annulata*, more commonly recognized as the kiara fig, the distinguished *Swietenia mahagoni*, widely acknowledged as mahogany, and lastly, *Gnetum gnemon*, commonly referred to as melinjo. The findings pertaining to the botanical examination within the arboreal classification in the vicinity of indigenous colonies of stingless bees are duly illustrated in Figure 2.

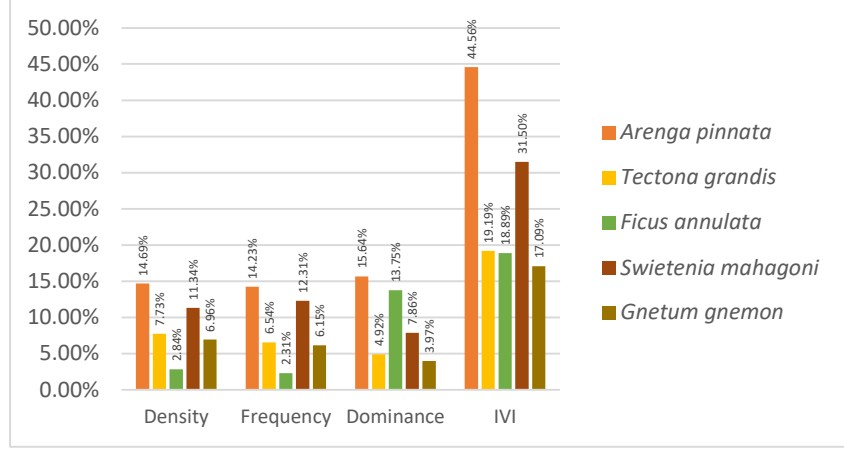

**Figure 2.** Relative density, relative frequency, relative dominance, and importance value index in the tree category in Sumedang Regency.

According to Figure 2, there are noticeable variances between each species' relative density levels. The total number of members of each species found throughout the entire plot area is represented by relative density. With a relative density of 14.69%, the Aren palm (*Arenga pinnata*) species has the highest value in the tree category. It is followed by the mahogany (*Swietenia mahagoni*), teak (*Tectona grandis*), melinjo (*Gnetum gnemon*), and kiara (*Ficus annulata*) species.

The tree species with the highest relative frequency is Aren (*Arenga pinnata*), followed by mahogany (*Swietenia mahagoni*), teak (*Tectona grandis*), melinjo (*Gnetum gnemon*), and kiara (*Ficus annulata*) at 12.31%, 6.54%, 6.96%, and 2.84%, respectively. The tree species with the highest relative dominance value is Aren (*Arenga pinnata*), followed by kiara (*Ficus annulata*), mahogany (*Swietenia mahagoni*), teak (*Tectona grandis*), and melinjo (*Gnetum gnemon*) at 13.75%, 7.86%, 4.92%, and 3.97%, respectively. Aren (*Arenga pinnata*) has the greatest importance value index (IVI), followed by mahogany (*Swietenia mahagoni*), teak (*Tectona grandis*), kiara (*Ficus annulata*), and melinjo (*Gnetum gnemon*) at 31.50%, 19.19%, 18.89%, and 17.09%. The Aren (*Arenga pinnata*) species has the highest overall values for RD, RF, RD, and IVI in the tree category with an RD of 14.69%, an RF of 14.23%, an RD of 15.64%, and an IVI of 44.56%.

Different plants predominate in each sub-district. By examining the IVI value, it is possible to identify the predominant plant species in a given area. When relative density, relative frequency, and relative dominance are added up, the result is the IVI, which is a measurement of how dominating a species is in its ecosystem. Table 2 shows the prevalent plant information for each sub-district.

**Table 2.** Comparison of dominant tree species by sub-district.

| No. | Sub District | Dominant Tree | Family | INP |
|---|---|---|---|---|
| 1 | Tanjungkerta | Jati (*Tectona grandis* L.f.) | Lamiaceae | 52.89% |
| | | Kelapa (*Cocos nucifera* L.) | Arecaceae | 49.80% |
| | | Sengon (*Albizia chinensis* (Osbeck) Merr.) | Fabaceae | 35.43% |
| | | Mahoni (*Swietenia mahagoni* (L.) Jacq.) | Meliaceae | 28.54% |
| | | Aren (*Arenga pinnata* (Wurmb) Merr.) | Arecaceae | 22.27% |
| 2 | Cisitu | Aren (*Arenga pinnata* (Wurmb) Merr.) | Arecaceae | 62.38% |
| | | Mahoni (*Swietenia mahagoni* (L.) Jacq.) | Meliaceae | 46.91% |
| | | Sobsis (*Maesopsis eminii* Engl.) | Rhamnaceae | 29.77% |
| | | Mangga (*Mangifera indica* L.) | Anacardiaceae | 21.93% |
| | | Cengkeh (*Syzygium aromaticum* (L.) Merr. & L.M.Perry) | Myrtaceae | 18.79% |
| 3 | Darmaraja | Aren (*Arenga pinnata* (Wurmb) Merr.) | Arecaceae | 86.61% |
| | | Kiara (*Ficus annulata* Blume) | Moraceae | 59.51% |
| | | Sempur (*Dillenia obovata* (Blume) Hoogland) | Dilleniaceae | 34.55% |
| | | Tisuk (*Hibiscus macrophyllus* Roxb. ex Hornem.) | Malvaceae | 30.93% |
| | | Kedoya (*Dysoxylum gaudichaudianum* Miq.) | Meliaceae | 16.19% |
| 4 | Buah Dua | Melinjo (*Gnetum gnemon* L.) | Gnetaceae | 74.74% |
| | | Mahoni (*Swietenia mahagoni* (L.) Jacq.) | Meliaceae | 32.81% |
| | | Nangsi (*Oreocnide rubescens* (Blume) Miq.) | Urticaceae | 30.07% |
| | | Kiara (*Ficus annulata* Blume) | Moraceae | 27.28% |
| | | Aren (*Arenga pinnata* (Wurmb) Merr.) | Arecaceae | 14.78% |

As seen in Table 1 above, the Aren tree may be found in every sub-district. The family Arecaceae includes the Aren tree (*Arenga pinnata*). Aren is a plant that can thrive in a variety of soil types, including clay, limestone, and sand. However, overly acidic soils are not suitable for this plant's growth. The delicious liquid called "nira", which may be discovered in the flower stem of Aren, gives the plant its economic value. Its fruit can also be used to

make kolang kaling. As a result, the neighborhood has transformed Aren into a farmland crop [22]. Bees can find food in the nectar and pollen found in its blossoms. Because Aren trees contain nira, a pleasant liquid, honeybees are drawn to them. Stingless bees can find food and places to nest in the research area's abundance of Aren trees. Compared to dead trees or other objects, these bees tend to favor living trees as nesting places [23].

3.1.2. Vertical Plants

A total of 77 species of columnar plants from 29 families were discovered using the vegetation analysis. Awi bitung (*Dendrocalamus asper*), awi gombong (*Gigantochloa pseudoarundinacea*), mahogany (*Swietenia mahagoni*), banana (*Musa paradisiaca*), and tisuk (*Hibiscus macrophyllus*) are the five species that predominate in the category of columnar plants. Figure 3 displays the findings of the vegetation study for the category of columnar plants around the stingless bee nests.

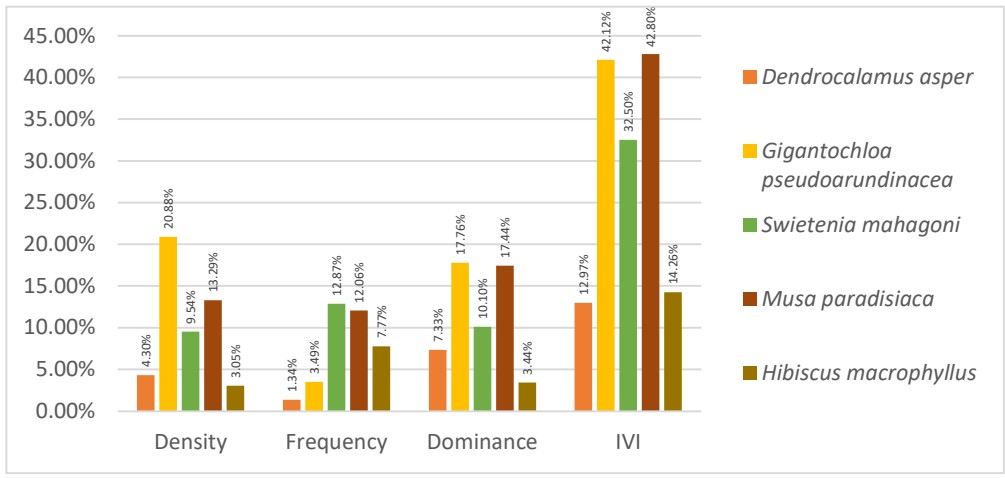

**Figure 3.** Relative density, relative frequency, relative dominance, and importance value index in the columnar plant category in Sumedang Regency.

According to Figure 3, the species *Gigantochloa pseudoarundinacea* had the highest relative density in the category of columnar plants, with a relative density of 20.88%. This species was followed by *Musa paradisiaca* at 13.29%, *Swietenia mahagoni* at 9.54%, *Dendrocalamus asper* at 4.30%, and *Hibiscus macrophyllus* at 3.05%. *Swietenia mahagoni* had the highest relative frequency in the columnar plant category (12.87%), followed by *Musa paradisiaca* (12.06%), *Hibiscus macrophyllus* (7.77%), *Gigantochloa pseudoarundinacea* (3.49%), and *Dendrocalamus asper* (1.34%).

With a relative dominance of 17.76%, *Gigantochloa pseudoarundinacea* had the highest relative dominance in the category of columnar plants, followed by *Musa paradisiaca* (17.44%), *Swietenia mahagoni* (10.10%), *Dendrocalamus asper* (7.33%), and *Hibiscus macrophyllus* (3.44%). With an IVI of 42.80%, *Musa paradisiaca* had the highest importance value index (INP) among the columnar plant species, followed by *Gigantochloa pseudoarundinacea* (42.12), *Swietenia mahagoni* (32.50%), *Hibiscus macrophyllus* (14.26%), and *Dendrocalamus asper* (12.97%). Table 3 contains information on the main flora in each sub-district.

The banana (*Musa paradisiaca*) species was identified as having the greatest IVI from each sub-district, with an IVI value of 82.63% in the Tanjungkerta sub-district. The banana plant can be found all over Indonesia. Because it can survive in a variety of soil conditions, growing it is relatively simple. Only one fruit is produced by this plant before it dies; however, shoots from its stem will develop into banana offshoots [24]. Bananas are one of the food sources for stingless bees because this plant produces both nectar and pollen, according to a study by [25]. Bananas grow quite easily in a variety of soil types; thus, their availability makes them a major food source for stingless bees. According to [26], bees can use any flowering plant that produces nectar, pollen, or resin as a food source.

**Table 3.** Comparison of dominant trees in the columnar plant category by sub-district.

| No. | Sub District | Dominant Tree | Family | IVI |
|---|---|---|---|---|
| 1 | Tanjungkerta | Pisang (*Musa paradisiaca* L.) | Musaceae | 82.63% |
| | | Bambu (*Bambusa* sp.) | Poaceae | 27.49% |
| | | Mahoni (*Swietenia mahagoni* (L.) Jacq.) | Meliaceae | 23.36% |
| | | Jati (*Tectona grandis* L.f.) | Lamiaceae | 28.12% |
| | | Mangga (*Mangifera indica* L.) | Anacardiaceae | 17.30% |
| 2 | Cisitu | Mahoni (*Swietenia mahagoni* (L.) Jacq.) | Meliaceae | 57.67% |
| | | Pisang (*Musa paradisiaca* L.) | Musaceae | 38.92% |
| | | Awi gombong (*Gigantochloa verticillata* (Willd.) Munro) | Poaceae | 37.20% |
| | | Awi tali (*Gigantochloa apus* (Schult.f.) Kurz ex Munro) | Poaceae | 12.53% |
| | | Kopi (*Coffea* sp.) | Rubiaceae | 10.38% |
| 3 | Darmaraja | Kaliandra merah (*Calliandra houstoniana var. calothyrsus* (Meisn.) Barneby) | Fabaceae | 49.81% |
| | | Awi bitung (*Dendrocalamus asper* Backer ex K.Heyne) | Poaceae | 49.54% |
| | | Haur hejo (*Bambusa tuldoides*) | | 28.90% |
| | | Awi gombong (*Gigantochloa verticillata* (Willd.) Munro) | Poaceae | 28.02% |
| | | Waregu (*Rhapis excelsa* (Thunb.) A.Henry) | Arecaceae | 26.55% |
| 4 | Buah Dua | Melinjo (*Gnetum gnemon* L.) | Gnetaceae | 49.92% |
| | | Awi gombong (*Gigantochloa verticillata* (Willd.) Munro) | Poaceae | 38.62% |
| | | Langkap (*Arenga obtusifolia* Mart.) | Arecaceae | 36.05% |
| | | Mahoni (*Swietenia mahagoni* (L.) Jacq.) | Meliaceae | 28.08% |
| | | Nangsi (*Oreocnide rubescens* (Blume) Miq.) | Urticaceae | 25.05% |

Along with bananas, swietenia mahagoni (*Gigantochloa pseudoarundinacea*) and mahogany (*Swietenia mahagoni*) are other dominant species that may be found in practically all sub-districts. Mahogany is a plant that adapts well to its surroundings. It belongs to the class of plants that may grow in a variety of well-draining soil types and do not require a particular type of soil to thrive. As a result, mahogany is widely available around the world. *Tetragonula* bees consume the pollen and resin produced by the mahogany plant, which also produces seeds. These bees make their bitter honey by combining pollen and resin. Bitter honey does not have the characteristic honey aroma; instead, it smells more like medicine. Since it has less sugar than conventional honey, the bitter flavor is more pronounced. The blossoms of mahogany trees near bee nests are the source of bitter honey [27].

The lowland bamboo species known as gigantochloa (*Gigantochloa pseudoarundinacea*) flourishes in tropical and humid regions. In the Sumedang region, this plant is comparatively prevalent in agroforestry. The local population extensively cultivates it for a variety of uses, such as water pipes, traditional musical instruments, and building materials [28,29].

### 3.1.3. Saplings

In total, 106 sapling species from 33 families were identified using the vegetation analysis. The gigantochloa bamboo (*Gigantochloa pseudoarundinacea*), awi tali bamboo (*Gigantochloa apus*), common bamboo (*Bambusa* sp.), Bambusa tuldoides, and red calliandra (*Calliandra calothyrsus*) are the five plant species that make up the majority of the saplings. Figure 4 displays the findings of the vegetation analysis for the sapling category near the stingless bee nests.

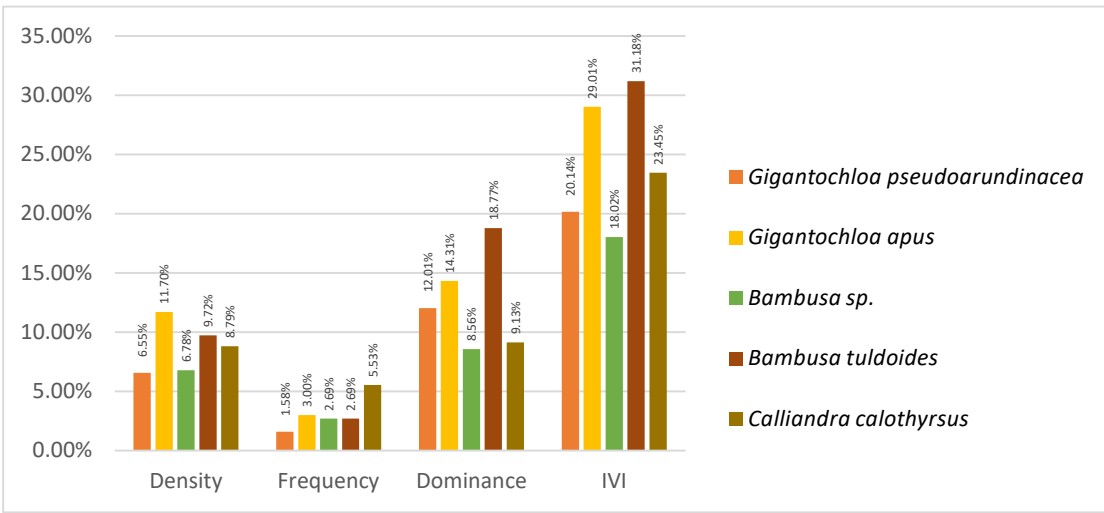

**Figure 4.** Relative density, relative frequency, relative dominance, and importance value index in the sapling category in Sumedang Regency.

According to the data presented in Figure 4, it is evident that the species exhibiting the highest relative density value within the sapling category is *awi tali* bamboo, scientifically known as *Gigantochloa apus*, boasting an impressive value of 11.70%. Following closely behind is *Bambusa tuldoides*, with a respectable relative density value of 9.72%. Notably, the red calliandra, or *Calliandra calothyrsus*, also makes a noteworthy appearance with a relative density value of 8.79%. Additionally, common bamboo, scientifically classified as *Bambusa* sp., exhibits a relative density value of 6.78%, while gigantochloa bamboo, known as *Gigantochloa pseudoarundinacea*, showcases a relative density value of 6.55%.

The sapling category exhibits a notable relative frequency value, with the red calliandra species (*Calliandra calothyrsus*) taking the lead at an impressive 5.53%. Following closely behind is awi tali bamboo (*Gigantochloa apus*) at a respectable 3.00%. Noteworthy contributions are also observed from *Bambusa tuldoides* and common bamboo (*Bambusa* sp.), both exhibiting a relative frequency value of 2.69%. Lastly, gigantochloa bamboo (*Gigantochloa pseudoarundinacea*) makes a modest appearance with a relative frequency value of 1.58%.

The sapling category exhibits the utmost relative dominance value in *Bambusa tuldoides*, with an impressive value of 18.77%. Subsequently, awi tali bamboo (*Gigantochloa apus*) follows suit with a respectable value of 14.31%. Gigantochloa bamboo (*Gigantochloa pseudoarundinacea*) demonstrates a noteworthy relative dominance value of 12.01%. Moreover, red calliandra (*Calliandra calothyrsus*) showcases a commendable value of 9.13%. Lastly, common bamboo (*Bambusa* sp.) concludes this list with a noteworthy relative dominance value of 8.56%.

The sapling category exhibits a distribution of the importance value index (IVI), with *Bambusa tuldoides* taking the lead with an impressive value of 31.18%. Following closely behind is the awi tali bamboo, scientifically known as *Gigantochloa apus*, with a noteworthy IVI of 29.01%. Notably, red calliandra, or *Calliandra calothyrsus*, secures a respectable IVI of 23.45%. Additionally, gigantochloa bamboo, scientifically referred to as *Gigantochloa pseudoarundinacea*, demonstrates a noteworthy presence with an IVI of 20.14%. Lastly, common bamboo, identified as *Bambusa* sp., exhibits a commendable IVI of 18.02%. A comprehensive compilation of the data pertaining to the dominant plant species within each sub-district is provided in Table 4.

**Table 4.** Comparison of dominant trees in the sapling category by sub-district.

| No. | Sub District | Dominant Tree | Family | IVI |
|---|---|---|---|---|
| 1 | Tanjungkerta | Bambu (*Bambusa* sp.) | Poaceae | 38.65% |
|  |  | Pinang (*Areca catechu* L.) | Arecaceae | 25.03% |
|  |  | Tisuk (*Hibiscus macrophyllus* Roxb. ex Hornem.) | Malvaceae | 23.22% |
|  |  | Pisang (*Musa paradisiaca* L.) | Musaceae | 22.67% |
|  |  | Kopi (*Coffea* sp.) | Rubiaceae | 16.89% |
| 2 | Cisitu | Awi tali (*Gigantochloa apus* (Schult.f.) Kurz ex Munro) | Poaceae | 48.06% |
|  |  | Kopi robusta (*Coffea canephora* Pierre ex A.Froehner) | Rubiaceae | 25.01% |
|  |  | Haur hejo (*Bambusa tuldoides* Munro) | Poaceae | 22.42% |
|  |  | Ki sereuh (*Piper aduncum* L.) | Piperaceae | 20.81% |
|  |  | Kopi (*Coffea* sp.) | Rubiaceae | 19.80% |
| 3 | Darmaraja | Haur hejo (*Bambusa tuldoides* Munro) | Poaceae | 72.53% |
|  |  | Kaliandra merah (*Calliandra houstoniana var. calothyrsus* (Meisn.) Barneby) | Fabaceae | 62.66% |
|  |  | Awi gombong (*Gigantochloa verticillata* (Willd.) Munro) | Poaceae | 36.42% |
|  |  | Ki seueur (*Antidesma velutinosum* Blume) | Phyllanthaceae | 22.12% |
|  |  | Waregu (*Rhapis excelsa* (Thunb.) A.Henry) | Arecaceae | 15.86% |
| 4 | Buah Dua | Awi tali (*Gigantochloa apus* (Schult.f.) Kurz ex Munro) | Poaceae | 65.39% |
|  |  | Bambu (*Bambusa* sp.) | Poaceae | 40.59% |
|  |  | Rotan (*Calamus* sp.) | Arecaceae | 31.05% |
|  |  | Awi tamiyang (*Schizostachyum blumei* Nees) | Poaceae | 16.61% |
|  |  | Awi bitung (*Dendrocalamus asper* Backer ex K.Heyne) | Poaceae | 13.75% |

In every sub-district, the *Bambusa tuldoides* species exhibited the most elevated importance value index (IVI). Specifically, this species demonstrated its prominence within the Darmaraja sub-district, boasting an IVI value of 72.53%. *Bambusa tuldoides*, colloquially referred to as "haur hejo", represents a taxonomically classified bamboo species within the *Poaceae* botanical family. The indigenous range of this particular species is primarily confined to the vast expanse of China, with additional occurrences documented in various regions of Southeast Asia as well as other countries characterized by tropical and subtropical climates [30]. This particular species of bamboo is commonly utilized for the creation of artisanal handicrafts, showcasing its versatility and cultural significance. Additionally, it is worth noting that the tender shoots of this bamboo variety hold culinary value and can be incorporated into various gastronomic delights [31].

The presence of bamboo from the Bambusa genus can be observed in every sub-district, without exception. Bamboo, a botanical entity of considerable interest, manifests as a clumping plant, characterized by the presence of multiple stems that exhibit a gradual growth pattern over time. Bamboo, a remarkably versatile botanical specimen, exhibits a good capacity for acclimatization, demonstrating its ability to flourish in diverse ecological settings in both arid and humid terrains. Furthermore, this resilient plant showcases its adaptability by thriving in both low-lying regions and elevated altitudes [32].

### 3.1.4. Seedling

Seedlings, the nascent individuals of plants, represent a pivotal stage in the life cycle of various botanical species. Upon careful examination of the outcomes derived from the careful analysis of the vegetation, a total of 103 distinct species of seedlings belonging to 38 diverse families have been identified. In the realm of seedlings, it is noteworthy to acknowledge the prevalence of five prominent plant species, namely, *Calliandra calothyrsus*, *Coffea robusta*, *Swietenia mahagoni*, *Musa paradisiaca*, and *Hibiscus macrophyllus*. The findings

pertaining to the seedling category vegetation analysis in the vicinity of the stingless bee habitat are illustrated in Figure 5.

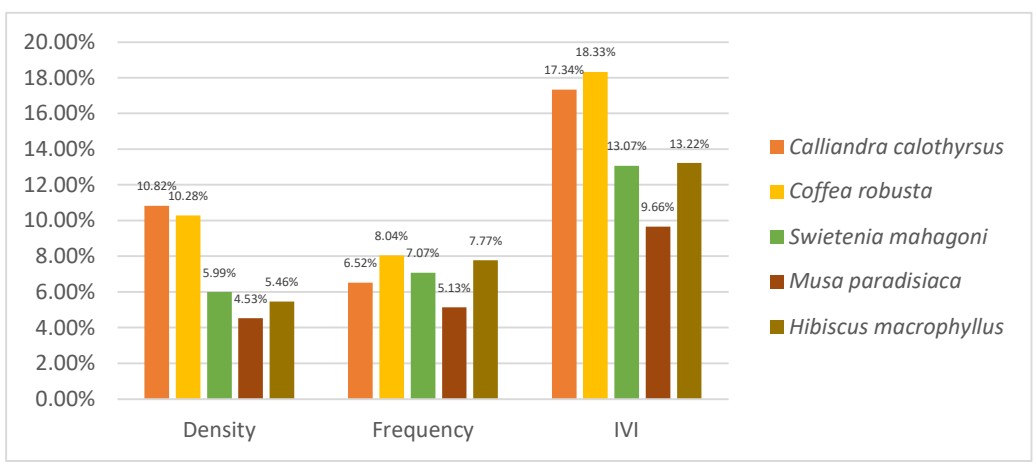

**Figure 5.** Relative density, relative frequency, relative dominance, and importance value index in the seedling category in Sumedang Regency.

According to the data presented in Figure 5, it is evident that the species *Calliandra calothyrsus* exhibited the highest relative density value within the seedling category, amounting to an impressive 10.82%. Following closely behind, we observe *Coffea robusta* with a relative density of 10.28%. Other notable species include *Swietenia mahagoni* with a relative density of 5.99%, *Hibiscus macrophyllus* with 5.46%, and *Musa paradisiaca* with 4.53%. These findings shed light on the distribution patterns and relative abundance of these species within the seedling category.

The seedling category exhibited a notable distribution of relative frequency values. Remarkably, *Coffea robusta* emerged as the frontrunner with an impressive relative frequency of 8.04%. Following closely behind, *Hibiscus macrophyllus* demonstrated a commendable relative frequency of 7.77%. Equally noteworthy, *Swietenia mahagoni* exhibited a respectable relative frequency of 7.07%. Additionally, *Calliandra calothyrsus* showcased a relative frequency of 6.52%. Lastly, *Musa paradisiaca*, while not reaching the same heights as its counterparts, still displayed a respectable relative frequency of 5.13%.

The seedling category exhibited a vast distribution of the importance value index (IVI), with *Coffea robusta* taking the lead with an impressive IVI of 18.33%. Following closely behind is *Calliandra calothyrsus* with a respectable IVI of 17.34%. Notably, *Hibiscus macrophyllus* displayed a commendable IVI of 13.22%, while *Swietenia mahagoni* and *Musa paradisiaca* demonstrated IVIs of 13.07% and 9.66%, respectively.

The data pertaining to the prevailing flora in each sub-district can be readily observed in Table 5. Upon careful examination and analysis of the vegetation data, a total of 103 distinct species of seedlings belonging to 38 diverse families have been identified. In the realm of seedlings, it is noteworthy to mention that there exist five prominent plant species that command our attention: red calliandra (*Calliandra calothyrsus*), Robusta coffee (*Coffea robusta*), mahogany (*Swietenia mahagoni*), banana (*Musa paradisiaca*), and tisuk (*Hibiscus macrophyllus*). The findings pertaining to the seedling category vegetation analysis in the vicinity of the stingless bee habitat are illustrated in Figure 5.

**Table 5.** Comparison of dominant trees in the prevailing category by sub-district.

| No. | Sub District | Dominant Tree | Family | IVI |
|---|---|---|---|---|
| 1 | Tanjungkerta | Pisang (*Musa paradisiaca* L.) | Musaceae | 23.63% |
| | | Tisuk (*Hibiscus macrophyllus* Roxb. ex Hornem.) | Malvaceae | 18.67% |
| | | Ki ciat (*Ficus septica* Burm.f.) | Moraceae | 18.45% |
| | | Singkong (*Manihot esculenta* Crantz) | Euphorbiaceae | 12.97% |
| | | Jambu batu (*Psidium guajava* L.) | Myrtaceae | 11.10% |
| 2 | Cisitu | Kopi robusta (*Coffea canephora* Pierre ex A.Froehner) | Rubiaceae | 38.98% |
| | | Mahoni (*Swietenia mahagoni* (L.) Jacq.) | Meliaceae | 27.47% |
| | | Kaliandra merah (*Calliandra houstoniana* var. *calothyrsus* (Meisn.) Barneby) | Fabaceae | 16.46% |
| | | Tisuk (*Hibiscus macrophyllus* Roxb. ex Hornem.) | Malvaceae | 11.25% |
| | | Singkong (*Manihot esculenta* Crantz) | Euphorbiaceae | 6.63% |
| 3 | Darmaraja | Kaliandra merah (*Calliandra houstoniana* var. *calothyrsus* (Meisn.) Barneby) | Fabaceae | 45.05% |
| | | Waregu (*Rhapis excelsa* (Thunb.) A.Henry) | Arecaceae | 17.17% |
| | | Kopi robusta (*Coffea canephora* Pierre ex A.Froehner) | Rubiaceae | 15.84% |
| | | Ki sereuh (*Piper aduncum* L.) | Piperaceae | 15.73% |
| | | Kedoya (*Dysoxylum gaudichaudianum* Miq.) | Meliaceae | 14.09% |
| 4 | Buah Dua | Mahoni (*Swietenia mahagoni* (L.) Jacq.) | Meliaceae | 25.25% |
| | | Rotan (*Calamus* sp.) | Arecaceae | 20.17% |
| | | Langkap (*Arenga obtusifolia* Mart.) | Arecaceae | 19.40% |
| | | Ki sereuh (*Piper aduncum* L.) | Piperaceae | 15.22% |
| | | Mara (*Macaranga* sp.) | Euphorbiaceae | 14.50% |

In every sub-district under investigation, the species that exhibited the most elevated INP (Index of Naturalness and Purity) was *Calliandra calothyrsus*, specifically observed in the Darmaraja sub-district. Remarkably, this particular species demonstrated an IVI value of 45.05%, signifying a substantial level of naturalness and purity within its ecological niche. *Calliandra calothyrsus*, a member of the *Fabaceae* family, is a botanical entity that warrants scholarly attention. This particular botanical specimen is renowned for its capacity for accelerated vegetative development, enabling it to attain vertical dimensions ranging from 2.5 to 3.5 m. The distribution of this particular species is frequently observed in the geographical region of Southeast Asia, as well as in various other tropical nations [33]. *Calliandra calothyrsus*, a botanical specimen of considerable renown, is widely acknowledged within the scientific community for its versatility as a plant species. It has garnered significant popularity owing to its inherent capacity for easy cultivation and its ability to regenerate vigorously subsequent to pruning. Moreover, it exhibits the ability to bloom incessantly throughout the entire duration of the year, thereby bestowing upon it a profound significance in the realm of honey production [34].

Furthermore, it is worth noting the presence of the *Coffea robusta*, a species of considerable significance, known to inhabit the geographical regions of Cisitu and Darmaraja sub-districts. *Coffea robusta*, a plantation crop, is a member of the Rubiaceae family. As per the scholarly investigation conducted by [35], it has been observed that *Coffea robusta* exhibits superior growth adaptability in comparison to its counterparts within the realm of coffee species. The cultivation of this particular species is extensively practiced by various communities due to its significant economic worth. It has been observed, as elucidated by [36], that *Coffea robusta* demonstrates a superior level of resistance when confronted with debilitating leaf rust disease in comparison to its counterparts within the coffee species. The existence of bees within coffee plantations engenders a harmonious mutualistic symbiosis, wherein both the bees and coffee plants derive substantial benefits from their interdependence. As per the research conducted by [37], it has been observed that the symbiotic

relationship between bees and coffee plants exhibits a notable augmentation in coffee bean yield, approximately amounting to a 22% increase. Moreover, this harmonious association also yields honey characterized by a sucrose content of 28%. According to a study by [38], in the Costa Rican pollen spectrum of *Tetragonisca angustula* honey, the dominance of *Coffea arabica* pollen (54.3%) conferred the coffee a unifloral attribute. Therefore, an important contribution to the chemical quality and bioactive properties of *Tetragonisca angustula* honey will derive from coffee nectar.

3.1.5. Understory Plants

The topic of understory plants is indeed a fascinating subject within the realm of biodiversity. According to the findings derived from the vegetation analysis, a total of 82 distinct species belonging to 35 diverse families were successfully identified within the understory plant community. In the realm of the understory plant classification, it is noteworthy to highlight the presence of the five most prominent plant species: *Achyranthes aspera*, *Axonopus compressus*, *Justicia* sp., *Oplismenus* sp. and *Synedrella nodiflora*. The findings pertaining to the examination of vegetation within the understory plant classification in the vicinity of the nesting sites of untamed *Tetragonula laeviceps* bees are illustrated in Figure 6.

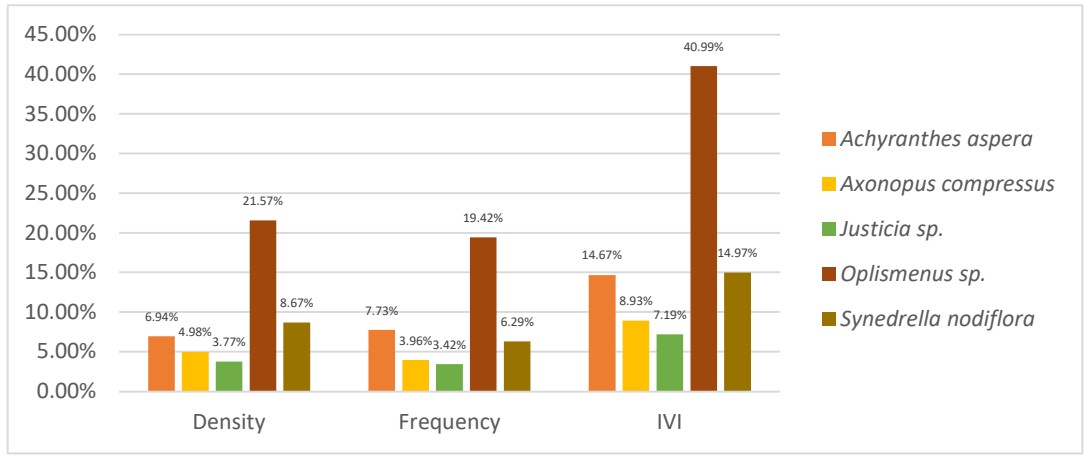

**Figure 6.** Relative density, relative frequency, relative dominance, and importance value index in the understory plan category in Sumedang Regency.

As per the findings depicted in Figure 6, the species *Oplismenus* sp. exhibited the most substantial relative density within the understory plant category, amounting to an impressive 21.57%. Subsequent to the aforementioned taxon, *Synedrella nodiflora* emerged with a relative density of 8.67%. *Achyranthes aspera*, *Axonopus compressus*, and *Justicia* sp. exhibited relative densities of 6.94%, 4.98%, and 3.77%, respectively.

The most notable observation in the understory plant category pertains to the species *Oplismenus* sp., which exhibited a relative frequency of 19.42%. Subsequently, we observed *Achyranthes aspera* with a relative frequency of 7.73%, *Synedrella nodiflora* with 6.29%, *Axonopus compressus* with 3.96%, and Justicia sp. with 3.42%.

The species *Oplismenus* sp. exhibited the most noteworthy IVI value within the understory plant category, reaching an impressive 40.99%. Following closely behind, we observed *Synedrella nodiflora* with a respectable IVI of 14.97%, *Achyranthes aspera* with a commendable 14.67%, *Axonopus compressus* with a noteworthy 8.93%, and *Justicia* sp. with a modest 7.19%. The comprehensive compilation of data pertaining to the dominant flora within each sub-district is presented in Table 6.

**Table 6.** Comparison of dominant trees in the understory category by sub-district.

| No. | Sub District | Dominant Tree | Family | IVI |
|-----|--------------|---------------|--------|-----|
| 1 | Tanjungkerta | Jotang kuda (*Synedrella nodiflora* (L.) Gaertn.) | Asteraceae | 28.63% |
| | | Jarong (*Achyranthes aspera* L.) | Amaranthaceae | 20.82% |
| | | Rumput pait (*Axonopus compressus* (Sw.) P.Beauv.) | Poaceae | 18.44% |
| | | Oplismenus (*Oplismenus* sp.) | Poaceae | 18.34% |
| | | Gandarusa (*Justicia* sp.) | Acanthaceae | 17.45% |
| 2 | Cisitu | Oplismenus (*Oplismenus* sp.) | Poaceae | 43.06% |
| | | Singonium (*Syngonium podophyllum* Schott) | Araceae | 16.97% |
| | | Jotang kuda (*Synedrella nodiflora* (L.) Gaertn.) | Asteraceae | 11.97% |
| | | Rumput israel (*Asystasia gangetica* (L.) T.Anderson) | Acanthaceae | 9.21% |
| | | Jarong (*Achyranthes aspera* L.) | Amaranthaceae | 8.05% |
| 3 | Darmaraja | Oplismenus (*Oplismenus* sp.) | Poaceae | 68.80% |
| | | Oplismenus (*Oplismenus undulatifolius* (Ard.) P.Beauv.) | Poaceae | 42.84% |
| | | Harendong (*Miconia crenata* (Vahl) Michelang.) | Melastomataceae | 10.70% |
| | | Jarong (*Achyranthes aspera* L.) | Amaranthaceae | 10.47% |
| | | Sembung rambat (*Mikania micrantha* Kunth) | Asteraceae | 7.74% |
| 4 | Buah Dua | Oplismenus (*Oplismenus* sp.) | Poaceae | 64.90% |
| | | Jarong (*Achyranthes aspera* L.) | Amaranthaceae | 18.88% |
| | | Rumput knop (*Hyptis capitata* Jacq.) | Lamiaceae | 12.99% |
| | | Gadung (*Dioscorea hispida* Dennst.) | Dioscoreaceae | 12.49% |
| | | Paku hata (*Lygodium* sp.) | Schizaeaceae | 11.84% |

Upon careful examination of the aforementioned table, it becomes apparent that the Oplismenus grass species exhibits a widespread distribution across all sub-districts under consideration. Furthermore, it is noteworthy that this particular species attains the highest Index of Natural Presence (IVI) within the confines of the Buah Dua sub-district. Oplismenus, scientifically known as *Oplismenus* sp., represents a botanical specimen that falls within the taxonomic classification of the Poaceae family. Its distribution covers diverse geographical regions, including, but not limited to, South Asia, East Asia, Southeast Asia, Australia, and South Africa. This particular botanical specimen exhibits a wide distribution range, primarily observed within the confines of tropical and subtropical regions. As per the research conducted by [39], it has been observed that Oplismenus grass exhibits a remarkable propensity for swift proliferation within forested regions characterized by the presence of arboreal flora, thereby establishing its dominance over other herbaceous species. Hence, this particular species of grass possesses the inherent capability to exert dominance over the designated areas of observation.

*3.2. Comparison with Other Studies*

Although there are some studies about vegetation diversity and bees, most of these are qualitative studies and/or conducted in different parts of Indonesia. A study by [7] conducted in West Seram Regency, Maluku (a district in the eastern part of Indonesia) identifies several plant species that possess significant value as nutritional resources for *Apis* bees: avocado (*Persea americana*), palm sugar (Arenga pinnata), tamarind (*Tamarindus indica*), starfruit (*Averrhoa carambola*), sunflower (*Helianthus annuus*), chili (*Capsicum frutescens*), pomegranate (*Punica granatum*), langsat (*Lansium domesticum*), durian (*Durio zibethinus*), gamal (*Gliricidia sepium*), corn (*Zea mays*), guava (*Psidium guajava*), water apple (*Syzygium aqueum*), cashew (*Anacardium occidentale*), Java apple (*Syzygium malaccense*), pomelo (*Citrus maxima*), teak (*Tectona grandis*), and coffee (*Coffea* sp.). From these findings, only three plant species, palm sugar (*Arenga pinnata*), Java apple (*Syzygium malaccense*), and coffee (*Coffea*

sp.) were also found in our study. The difference in plant diversity might arise from the difference in the micro-climate between Maluku and Sumedang, with Maluku being much drier and having a strong wind compared to Sumedang [40].

Another study conducted in Bengkayang Regency (West Kalimantan) found a number of species. There are 23 species of tree-level vegetation dominated by the manja species (*Xanthophyllim amoenum*), 10 species of pole-level vegetation dominated by the kurpa species (*Lepisanthes tetraphylla*), 5 species of sapling-level vegetation dominated by the jambu (*Syzygium chloranthum*), and 5 species of seedling-level vegetation dominated by jambu perancis species (*Bellucia pentamera*). The most frequently encountered plants serving as a food source belong to the Myrtaceae family, including jambu, kurpa, and pohon pelawan (*Tristaniopsis merguensis*). Other species have been recognized as prospective food resources for stingless bees, such as durian (*Durio zibethinus*), rambutan (*Nephelium lappaceum*), water apple (*Syzygium aqueum*), and eggplant (*Solanum melongena*) [41]. However, none of these species were found in this study's research area. Although these species can also be found in Sumedang, one of the reasons that might cause this difference is that the study by Sanjaya was conducted in mixed gardens compared to forests as the research area of this study.

One study, which was the closest to this study in terms of the study area (Sumedang) and the type of bees (*Tetragonula*), was conducted by [42] and found the following plant species: pineapples (*Ananas comosus*), cleome (*Cleome rutidospermae*), Eleusineum grass (*Pennisetum plystachion*), huperzia (*Huperzia* sp.), eleusine grass (*Eleusine indica*), and ageratum (*Ageratum conyzoides*). However, two main differences between this study and that of Rismayanti's are that the latter was conducted in areas that were located in close proximity to human settlements and that the stingless bees studied were cultivated bees instead of wild bees, causing these studies to yield different results. Additionally, as the scope of the vegetation structure of the qualitative study conducted by [42] was much smaller than this quantitative study, a conclusive comparison could not be done.

With the highest number of sampling areas and the most extensive vegetation structures observed compared to other studies about vegetation diversity and bees, this research provides a much clearer and more comprehensive description of the existing conditions in Sumedang Regency forest areas. Additionally, this study provides additional value because it quantifies the data compared to the other qualitative research.

## 4. Conclusions

The subject matter at hand pertains to the noteworthy richness of vegetation diversity. The assemblage of 52 distinct tree species, 77 pole species, 107 sapling species, 103 seedling species, and 82 ground cover (understory plants) species in the immediate vicinity of *Tetragonula laeviceps* bee nests in Sumedang Regency serves as a striking testament to the prodigious and multifaceted tapestry of plant diversity that thrives in this particular ecological setting. The preservation of biodiversity is of utmost importance in upholding the integrity of ecological systems and providing sustenance for a myriad of fauna species.

The arboreal classification exhibits prominence in terms of relative density (RD), relative frequency (RF), relative dominance (RD), and importance value index (IVI). The observed phenomenon can be primarily attributed to the prevalence of the Aren palm (*Arenga pinnata*) within this particular taxonomic classification. Indeed, the presence of palm trees in the local ecosystem plays a substantial role, thereby contributing significantly to the overall biodiversity. It is worth noting that these palms exhibit a capacity to influence the atmospheric environment by generating high Ice Nucleating Particle (INP) values, which have been quantified at an impressive 44.56%. This observation underscores the profound significance of these habitats in bolstering the vitality of bee populations and the overall well-being of the ecosystem.

The calculated Shannon Diversity Index (H') values for all vegetation categories surpass the threshold of 3, indicating a considerable magnitude of biodiversity within each respective category. This observation indicates that the ecological milieu surrounding the nests of *Tetragonula laeviceps* bees exhibits a notable diversity and robustness, wherein a

plethora of plant species harmoniously coexist. The phenomenon of high biodiversity has been widely observed to exhibit a positive correlation with the stability and overall health of ecosystems.

In summary, the vicinity surrounding the nests of *Tetragonula laeviceps* bees in Sumedang Regency exhibits a vast array of botanical diversity, wherein the arboreal taxa, particularly the Aren palm, assume a preeminent position. This vast diversity observed in various ecosystems serves as a pivotal factor in fostering a resilient ecological framework, as evidenced by the substantial values obtained from biodiversity indices. The comprehension and preservation of this botanical existence are of utmost importance, not solely for the sustenance of *Tetragonula laeviceps* bees, but also for the holistic maintenance of ecological equilibrium within the region.

**Author Contributions:** Conceptualization, S.W. and V.L.; methodology, S.W. and P.P.; software, V.L.; validation, S.W. and P.P.; formal analysis, F.R.; investigation, F.R.; resources, F.R.; data curation, F.R.; writing—original draft preparation, S.W.; writing—review and editing, P.P.; visualization, F.R.; supervision, P.P.; project administration, S.W.; funding acquisition, P.P. All authors have read and agreed to the published version of the manuscript.

**Funding:** This research was funded by "the Indonesia Ministry of Education and Culture through *Hibah Riset Kemitraan Dasar* (E-Asia Project) and *Penelitian Terapan Unggulan Perguruan Tinggi* (PTUPT), Riset Kompetensi Dosen Unpad, Academic Leadership Grant (ALG) Unpad number 1834/UN6.3.1/PT.00/2023" and "The APC was funded by Universitas Padjadjaran".

**Institutional Review Board Statement:** Not applicable.

**Data Availability Statement:** The data presented in this study are available on request from the corresponding author. The data are not publicly available due to that the data is primary data and can not be shared.

**Acknowledgments:** The authors would say thanks to all research schemes that fully supported this study: Ministry of Education and Culture of Indonesia through *Hibah Riset Penelitian Terapan Unggulan Perguruan Tinggi* (*PTUPT*) dan *Kemitraan Dasar*; *Hibah Riset* Academic Leadership Grant (ALG) and *Riset Kompetensi Dosen Unpad* from Universitas Padjadjaran. The author would also like to thank several parties who have participated and provided support during the preparation of the results of this study.

**Conflicts of Interest:** The authors declare no conflict of interest.

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
