# Peer review of "Vegetation Analysis of the Area Surrounding a Wild Nest of Stingless Bees Tetragonula laeviceps (Smith, 1857) in Sumedang Regency, West Java"

_diversity, doi:10.3390/d15111149_

Round 1

Reviewer 1 Report

Comments and Suggestions for Authors

Mérida, 27th September, 2023

Ms Diversity 2632441

Please, receive my comments for suggested adjustments if the authors consider them adequate.

Vegetation analysis of the area surrounding a wild nest of stingless bees (Tetragonula laeviceps, Smith 1857) in Sumedang Regency, West Java

Susanti Withaningsih, Valerie Lubay, Fakhru Rozi, and Parikesit Parikesit

Kindly suggest to Parikesit to repeat Parikesit, if possible, so the article can be cited with an initial. I know Indonesians do not use the name if it is the same as the surname, but in international journals would be respectful to have an initial, and not a void looking as if the initial was forgotten by the authors in the Reference list.

For example

Agussalim, A., Agus, A., Umami, N., & Budisatria, I. G. S. (2017). Variation of Honeybees Forages As Source of Nectar and Pollen Based on Altitude in Yogyakarta. Buletin Peternakan, 41(4), 448. https://doi.org/10.21059/buletinpeternak.v41i4.13593

The topic of the manuscript is of interest in bee science. There is a review of literature, a figure and Tables are adequate to show the location and results.

Title

This is correct Tetragonula laeviceps (Smith, 1857)

The parenthesis has a meaning in entomology, it is not an option.

Vegetation analysis of the area surrounding a wild nest of stingless bees Tetragonula laeviceps (Smith, 1857) in Sumedang Regency, West Java

General

Tetragonula always with Italics please. Other genera too.

Avoid excessive repeated adjectives (intricate, remarkable, esteemed, etc.)

Choose nest or hive, not both. Better Tetragonula laeviceps nests.

Abstract

There is no need to repeat Tetragonula laeviceps in the third line.

1.       Introduction

Only 5 references in the introduction seems not enough?

2.       Materials and Methods

The stingless bee Tetragonula laeviceps should have a voucher in a local entomological collection and an entomologist to backup its identification. This is easy to do, and will improve the scientific quality of this important study.

The plants listed in this manuscript should have a voucher in a local Herbarium with the corresponding botanical identifications, and the botanist responsible for that. Possibly the authors have done that in previous studies? Then. provide the name of the Herbarium, with the international code.

3.       Results and Discussion

Please, kindly check if the names of plant species and families need to be updated. I already suggested an expert.

Line 476. If useful, read this paper on Coffea arabica honey:

Moreno E, Vit P, Aguilar I, Barth OM. 2023. Melissopalynological spectrum of a Coffea arabica unifloral Tetragonisca angustula (Latreille, 1811) honey from Alajuela, Costa Rica. AIMS Agriculture and Food 8(3):804-831. https://www.aimspress.com/article/doi/10.3934/agrfood.2023043

Authors contributions

They were not informed.

Reference List

A Michener reference is missing.

Reviewer 2 Report

Comments and Suggestions for Authors

Line 64-65 repetition of sentence from l. 40-41.

Despite wast number of literature about pollination in the peper there are only 5 citations in „Introduction“ which chapter usually serves as starting point for topic of paper.

Abuse of some not frequent words as (meticulous/ly) – 11x, esteemed (16x).

l. 65, 130, 193, 231, 301, 350 etc. italics.

in chapter „results and discussion“ there is practically no comparison with numerous published results from papers dealing with vegetation diversity and bees. I recommend to add separate „discussion“ chapter and here to gather this info.

l. 175 „320 sampling points“ but on Fig. 1. there are 5 „sampling points“.

papers 11 and 28 are named „scholarly work“. Why? They are published normály in journals.

Lines 179 and following are identical to l. 220 and following.

p. 55 and following „exhibits a remarkable array of botanical diversity“. It is senseless without comparison with other published paper or own data from other environments.

Round 2

Reviewer 2 Report

Comments and Suggestions for Authors

Now improved text is much better than first version.

Author Response

Thank you very much for your valuable suggestions and comments on our manucript. Those comments are great assistance to me for improving and revising our manucript.

Thank you so much for your support to make this manucript can be publish in Diversity journal.